# SSE-PT: Sequential Recommendation Via Personalized Transformer

## Abstract

Temporal information is crucial for recommendation problems because user preferences are naturally dynamic in the real world. Recent advances in deep learning, especially the discovery of various attention mechanisms and newer architectures in addition to widely used RNN and CNN in natural language processing, have allowed for better use of the temporal ordering of items that each user has engaged with. In particular, the SASRec model, inspired by the popular Transformer model in natural languages processing, has achieved state-of-the-art results. However, SASRec, just like the original Transformer model, is inherently an un-personalized model and does not include personalized user embeddings. To overcome this limitation, we propose a Personalized Transformer (SSE-PT) model, outperforming SASRec by almost 5% in terms of NDCG@10 on 5 real-world datasets. Furthermore, after examining some random users' engagement history, we find our model not only more interpretable but also able to focus on recent engagement patterns for each user. Moreover, our SSE-PT model with a slight modification, which we call SSE-PT++, can handle extremely long sequences and outperform SASRec in ranking results with comparable training speed, striking a balance between performance and speed requirements. Our novel application of the Stochastic Shared Embeddings (SSE) regularization is essential to the success of personalization. Code and data are open-sourced at `https://github.com/SSE-PT/SSE-PT`.

## 1 Introduction

The sequential recommendation problem has been an important open research question, yet using temporal information to improve recommendation performance has proven to be challenging. SASRec, proposed by (Kang and McAuley, 2018) for sequential recommendation problems, has achieved state-of-the-art results and enjoyed more than 10x speed-up when compared to earlier CNN/RNN-based methods. However, the model used in SASRec is the standard Transformer which is inherently an un-personalized model. In practice, it is important to include a personalized Transformer in SASRec especially for recommender systems, but (Kang and McAuley, 2018) found that adding additional personalized embeddings did not improve the performance of their Transformer model, and postulate that the failure of adding personalization is due to the fact that they already use the user history and the user embeddings only contribute to overfitting. In this work, we propose a novel method, Personalized Transformer (SSE-PT), that successfully introduces personalization into self-attentive neural network architectures.

Introducing user embeddings into the standard transformer model is intrinsically difficult with existing regularization techniques, as unavoidably a large number of user parameters are introduced, which is often at the same scale of the number of training data. But we show that personalization can greatly improve ranking performance with a recent regularization technique called Stochastic Shared Embeddings (SSE) (Wu et al., 2019). The personalized Transformer (SSE-PT) model with SSE regularization works well for all 5 real-world datasets we consider without overfitting, outperforming previous state-of-the-art algorithm SASRec by almost 5% in terms of NDCG@10. Furthermore, after examining some random users' engagement history, we find our model is not only more interpretable but also able to focus on recent engagement patterns for each user. Moreover, our SSE-PT model with a slight modification, which we call SSE-PT++, can handle extremely long sequences and outperform SASRec in ranking results with comparable training speed, striking a balance between performance and speed requirements.

## 2 RELATED WORK

### 2.1 SESSION-BASED AND SEQUENTIAL RECOMMENDATION

Both session-based and sequential (i.e., next-basket) recommendation algorithms take advantage of additional temporal information to make better personalized recommendations. The main difference between session-based recommendations (Hidasi et al., 2015) and sequential recommendations (Kang and McAuley, 2018) is that the former assumes that the user ids are not recorded and therefore the length of engagement sequences are relatively short. Therefore, session-based recommendations normally do not consider user factors. On the other hand, sequential recommendation treats each sequence as a user's engagement history (Kang and McAuley, 2018). Both settings, do not explicitly require time-stamps: only the relative temporal orderings are assumed known (in contrast to, for example, timeSVD++ (Koren, 2009) using time-stamps). Initially, sequence data in temporal order are usually modelled with Markov models, in which a future observation is conditioned on the last few observed items (Rendle et al., 2010). In (Rendle et al., 2010), a personalized Markov model with user latent factors is proposed for more personalized results.

In recent years, deep learning techniques, borrowed from natural language processing (NLP) literature, are getting widely used in tackling sequential data. Like word sentences in NLP, item sequences in recommendations can be similarly modelled by recurrent neural networks (RNN) (Hidasi et al., 2015; Hidasi and Karatzoglou, 2018) and convolutional neural network (CNN) (Tang and Wang, 2018) models. Recently, attention models are increasingly used in both NLP (Vaswani et al., 2017; Devlin et al., 2018) and recommender systems (Liu et al., 2018; Kang and McAuley, 2018). SASRec (Kang and McAuley, 2018) is a recent method with state-of-the-art performance among the many deep learning models. Motivated by the Transformer model in neural machine translation (Vaswani et al., 2017), SASRec utilizes a similar architecture to the encoder part of the Transformer model. Our proposed model, SSE-PT, is a personalized extension of the transformer model.

### 2.2 REGULARIZATION TECHNIQUES

In deep learning, models with many more parameters than data points can easily overfit to the training data. This may prevent us from adding user embeddings as additional parameters into complicated models like the Transformer model (Kang and McAuley, 2018), which can easily have 20 layers with millions of parameters for a medium-sized dataset like Movielens10M (Harper and Konstan, 2016). $\ell_2$ regularization (Hoerl and Kennard, 1970) is the most widely used approach and has been used in many matrix factorization models in recommender systems; $\ell_1$ regularization (Tibshirani, 1996) is used when a sparse model is preferred. For deep neural networks, it has been shown that $\ell_p$ regularizations are often too weak, while dropout (Hinton et al., 2012; Srivastava et al., 2014) is more effective in practice. There are many other regularization techniques, including parameter sharing (Goodfellow et al., 2016), max-norm regularization (Srebro et al., 2005), gradient clipping (Pascanu et al., 2013), etc. Very recently, a new regularization technique called Stochastic Shared Embeddings (SSE) (Wu et al., 2019) is proposed as a new means of regularizing embedding layers. We find that the base version SSE-SE is essential to the success of our Personalized Transformer (SSE-PT) model.

## 3 METHODOLOGY

### 3.1 SEQUENTIAL RECOMMENDATION

Given $n$ users and each user engaging with a subset of $m$ items in a temporal order, the goal of sequential recommendation is to learn a good personalized ranking of top $K$ items out of total $m$ items for any given user at any given time point. We assume data in the format of $n$ item sequences:

$$s_i = (j_{i1}, j_{i2}, \ldots, j_{iT}) \text{ for } 1 \leq i \leq n. \tag{1}$$

Sequences $s_i$ of length $T$ contain indices of the last $T$ items that user $i$ has interacted with in the temporal order (from old to new). For different users, the sequence lengths can vary, but we can pad the shorter sequences so all of them have length $T$. We cannot simply randomly split data points into train/validation/test sets because they come in temporal orders. Instead, we need to make sure our training data is before validation data which is before test data temporally. We use last items in sequences as test sets, second-to-last items as validation sets and the rest as training sets. We

use ranking metrics such as NDCG@$K$ and Recall@$K$ for evaluations, which are defined in the Appendix.

### 3.2 PERSONALIZED TRANSFORMER ARCHITECTURE

Our model, which we call SSE-PT, is motivated by the Transformer model in (Vaswani et al., 2017) and (Kang and McAuley, 2018). It also utilizes a new regularization technique called stochastic shared embeddings (Wu et al., 2019). In the following sections, we are going to examine each important component of our Personalized Transformer (SSE-PT) model, especially the embedding layer, and the novel application of stochastic shared embeddings (SSE) regularization technique.

**Embedding Layer** We define a learnable user embedding look-up table $U \in R^{n \times d_u}$ and item embedding look-up table $V \in R^{m \times d_i}$, where $d_u$, $d_i$ are the number of hidden units for user and item respectively. We also specify learnable positional encoding table $P \in R^{T \times d}$, where $d = d_u + d_i$. So each input sequence $s_i \in R^T$ will be represented by the following embedding:

$$
E = \begin{bmatrix} [v_{j_{i1}}; u_i] + p_1 \\ [v_{j_{i2}}; u_i] + p_2 \\ \vdots \\ [v_{j_{iT}}; u_i] + p_T \end{bmatrix} \in R^{T \times d},
\tag{2}
$$

where $[v_{j_{it}}; u_i]$ represents concatenating item embedding $v_{j_{it}} \in R^{d_i}$ and user embedding $u_i \in R^{d_u}$ into embedding $E_t \in R^d$ for time $t$. Note that the main difference between our model and (Kang and McAuley, 2018) is that we introduce the user embeddings $u_i$, making our model personalized.

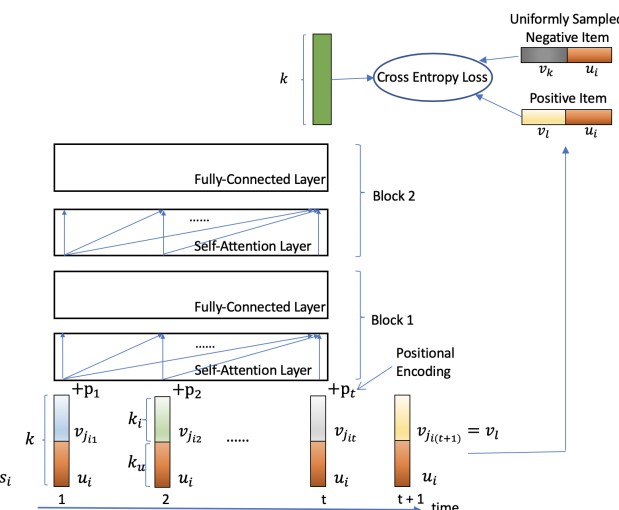

Figure 1: Illustration of our proposed SSE-PT model

**Transformer Encoder** On top of the embedding layer, we have $B$ blocks of self-attention layers and fully connected layers, where each layer extracts features for each time step based on the previous layer's outputs. Since this part is identical to the Transformer encoder used in the original papers (Vaswani et al., 2017; Kang and McAuley, 2018), we will skip the details.

**Prediction Layer** At time $t$, the predicted probability of user $i$ engaged item $l$ is:

$$
p_{itl} = \sigma(r_{itl}),
\tag{3}
$$

where $\sigma$ is the sigmoid function and $r_{itl}$ is the predicted score of item $l$ by user $l$ at time point $t$, defined as:

$$
r_{itl} = F_{t-1}^B \cdot [v_l; u_i],
\tag{4}
$$

where $F_{t-1}^B$ is the output hidden units associated with the transformer encoder at the last timestamp. Although we can use another set of user and item embedding look-up tables for the $u_i$ and $v_l$, we

find it better to use the same set of embedding look-up tables $U, V$ as in the embedding layer. But regularization for those embeddings can be different. To distinguish the $u_i$ and $v_l$ in (4) from $u_i, v_j$ in (2), we call embeddings in (4) output embeddings and those in (2) input embeddings.

The binary cross entropy loss between predicted probability for the positive item $l = j_{i(t+1)}$ and one uniformly sampled negative item $k \in \Omega$ is given as $-[\log(p_{itl}) + \log(1 - p_{itk})]$. Summing over $s_i$ and $t$, we obtain the objective function that we want to minimize is:

$$\sum_i \sum_{t=1}^{T-1} \sum_{k \in \Omega} -\big[\log(p_{itl}) + \log(1 - p_{itk})\big]. \tag{5}$$

At the inference time, top-$K$ recommendations for user $i$ at time $t$ can be made by sorting scores $r_{itl}$ for all items $\ell$ and recommending the first $K$ items in the sorted list.

**Novel Application of Stochastic Shared Embeddings**   The most important regularization technique to SSE-PT model is the Stochastic Shared Embeddings (SSE) (Wu et al., 2019). The main idea of SSE is to stochastically replace embeddings with another embedding with some pre-defined probability during SGD, which has the effect of regularizing the embedding layers. Without SSE, all the existing well-known regularization techniques like layer normalization, dropout and weight decay fail and cannot prevent the model from over-fitting badly after introducing user embeddings. (Wu et al., 2019) develops two versions of SSE, SSE-Graph and SSE-SE. In the simplest uniform case, SSE-SE replaces one embedding with another embedding uniformly with probability $p$, which is called SSE probability in (Wu et al., 2019). Since we don't have knowledge graphs for user or items, we simply apply the SSE-SE to our SSE-PT model. We find SSE-SE makes possible training this personalized model with $O(nd_u)$ additional parameters.

There are 3 different places in our model that SSE-SE can be applied. We can apply SSE-SE to input/output user embeddings, input item embeddings, and output item embeddings with probabilities $p_u$, $p_i$ and $p_y$ respectively. Note that input user embedding and output user embedding are always replaced at the same time with SSE probability $p_u$. Empirically, we find that SSE-SE to user embeddings and output item embeddings always helps, but SSE-SE to input item embeddings is only useful when the average sequence length is large, e.g., more than 100 in Movielens1M and Movielens10M datasets.

**Other Regularization Techniques**   Besides the *SSE* (Wu et al., 2019), we also utilized other widely used regularization techniques, including *layer normalization* (Ba et al., 2016), *batch normalization* (Ioffe and Szegedy, 2015), *residual connections* (He et al., 2016), *weight decay* (Krogh and Hertz, 1992), and *dropout* (Srivastava et al., 2014). Since they are used in the same way in the previous paper (Kang and McAuley, 2018), we omit the details to the Appendix.

### 3.3   Handling Long Sequences: SSE-PT++

To handle extremely long sequences, a slight modification can be made on the base SSE-PT model in terms of how input sequences $s_i$'s are fed into the SSE-PT neural network. We call the enhanced model SSE-PT++ to distinguish it from the previously discussed SSE-PT model, which cannot handle sequences longer than $T$.

The motivation of SSE-PT++ over SSE-PT comes from: sometimes we want to make use of extremely long sequences, $s_i = (j_{i1}, j_{i2}, \ldots, j_{it})$ for $1 \leq i \leq n$, where $t > T$, but our SSE-PT model can only handle sequences of maximum length of $T$. The simplest way is to sample starting index $1 \leq v \leq t$ uniformly and use $s_i = (j_{iv}, j_{i(v+1)}, \ldots, j_{iz})$, where $z = \min(t, v + T - 1)$. Although sampling the starting index uniformly from $[1, t]$ can accommodate long sequences of length $t > T$, this does not work well in practice. Uniform sampling does not take into account the importance of recent items in a long sequence. To solve this dilemma, we introduce an additional hyper-parameter $p_s$ which we call *sampling probability*. It implies that with probability $p_s$, we sample the starting index $v$ uniformly from $[1, t - T]$ and use sequence $s_i = (j_{iv}, j_{i(v+1)}, \ldots, j_{i(v+T-1)})$ as input. With probability $1 - p_s$, we simply use the recent $T$ items $(j_{i(t-T+1)}, \ldots, j_{it})$ as input. If the sequence $s_i$ is already shorter than $T$, then we always use the recent input sequence for user $i$.

Our proposed SSE-PT++ model can work almost as well as SSE-PT with a much smaller $T$. One can see in Table 2 with $T = 100$, SSE-PT++ can perform almost as well as SSE-PT. The time complexity

of the SSE-PT model is of order $O(T^2 d + T d^2)$. Therefore, reducing $T$ by one half would lead to a theoretically 4x speed-up in terms of the training and inference speeds. As to the model's space complexity, both SSE-PT and SSE-PT++ are of order $O(n d_u + m d_i + T d + d^2)$.

## 4 Experiments

In this section, we compare our proposed algorithms, Personalized Transformer (SSE-PT) and SSE-PT++, with other state-of-the-art algorithms on real-world datasets. We implement our codes in Tensorflow and conduct all our experiments on a server with 40-core Intel Xeon E5-2630 v4 @ 2.20GHz CPU, 256G RAM and Nvidia GTX 1080 GPUs.

**Datasets** We use 5 datasets. The first 4 have exactly the same train/dev/test splits as in (Kang and McAuley, 2018). The datasets are: *Beauty* and *Games* categories from Amazon product review datasets[1]; *Steam* dataset introduced in (Kang and McAuley, 2018), which contains reviews crawled from a large video game distribution platform; *Movielens1M* dataset (Harper and Konstan, 2016), a widely used benchmark datasets containing one million user movie ratings; *Movielens10M* dataset with ten million user ratings cleaned by us. Detailed dataset statistics are given in Table 4. One can easily see that the first 3 datasets have short sequences (average length < 12) while the last 2 datasets have very long sequences (> 10x longer).

**Evaluation Metrics** The evaluation metrics we use are standard ranking metrics, namely NDCG and Recall for top recommendations (See Appendix). We follow the same evaluation setting as the previous paper (Kang and McAuley, 2018): predicting ratings at time point $t + 1$ given the previous $t$ ratings. For a large dataset with numerous users and items, the evaluation procedure would be slow because (6) would require computing the ranking of all items based on their predicted scores for every single user. As a means of speed-up evaluations, we sample a fixed number $C$ (e.g., 100) of negative candidates while always keeping the positive item that we know the user will engage next. This way, both $R_{ij}$ and $\Pi_i$ will be narrowed down to a small set of item candidates, and prediction scores will only be computed for those items through a single forward pass of the neural network.

Ideally, we want both NDCG and Recall to be as close to 1 as possible, because NDCG@$K = 1$ means the positive item is always put on the top-1 position of the top-$K$ ranking list, and Recall@$K = 1$ means the positive item is always contained by the top-$K$ recommendations the model makes.

Table 1: Comparing various state-of-the-art temporal collaborative ranking algorithms on various datasets. The (A) to (E) are non-deep-learning methods, the (F) to (K) are deep-learning methods and the (L) to (O) are our variants. We did not report SSE-PT++ results for beauty, games and steam, as the input sequence lengths are very short (see Table 4), so there is no need for SSE-PT++.

| DATASET | BEAUTY | | GAMES | | STEAM | | ML-1M | |
|---|---|---|---|---|---|---|---|---|
| METRIC | RECALL@10 | NDCG@10 | RECALL@10 | NDCG@10 | RECALL@10 | NDCG@10 | RECALL@10 | NDCG@10 |
| (A) POPREC | 0.4003 | 0.2277 | 0.4724 | 0.2779 | 0.7172 | 0.4535 | 0.4329 | 0.2377 |
| (B) BPR | 0.3775 | 0.2183 | 0.4853 | 0.2875 | 0.7061 | 0.4436 | 0.5781 | 0.3287 |
| (C) FMC | 0.3771 | 0.2477 | 0.6358 | 0.4456 | 0.7731 | 0.5193 | 0.6983 | 0.4676 |
| (D) FPMC | 0.4310 | 0.2891 | 0.6802 | 0.4680 | 0.7710 | 0.5011 | 0.7599 | 0.5176 |
| (E) TRANSREC | 0.4607 | 0.3020 | 0.6838 | 0.4557 | 0.7624 | 0.4852 | 0.6413 | 0.3969 |
| (F) GRU4REC | 0.2125 | 0.1203 | 0.2938 | 0.1837 | 0.4190 | 0.2691 | 0.5581 | 0.3381 |
| (G) STAMP | 0.4607 | 0.3020 | 0.6838 | 0.4557 | 0.7624 | 0.4852 | 0.6413 | 0.3969 |
| (H) GRU4REC+ | 0.3949 | 0.2556 | 0.6599 | 0.4759 | 0.8018 | 0.5595 | 0.7501 | 0.5513 |
| (I) CASER | 0.4264 | 0.2547 | 0.5282 | 0.3214 | 0.7874 | 0.5381 | 0.7886 | 0.5538 |
| (J) SASREC | 0.4837 | 0.3220 | 0.7434 | 0.5401 | 0.8732 | 0.6293 | 0.8233 | 0.5936 |
| (K) HGN | 0.4469 | 0.2994 | 0.7164 | 0.5209 | 0.7426 | 0.4871 | 0.7584 | 0.5241 |
| (L) SSE-SASREC | 0.4878 | 0.3342 | 0.7517 | 0.5535 | 0.8697 | 0.6333 | 0.8230 | 0.5995 |
| (M) PT | 0.3954 | 0.2449 | 0.6427 | 0.4434 | 0.7535 | 0.4853 | 0.7658 | 0.5241 |
| (N) SSE-PT | **0.5028** | **0.3370** | **0.7757** | **0.5660** | **0.8772** | **0.6378** | 0.8341 | 0.6281 |
| (O) SSE-PT++ | – | – | – | – | – | – | **0.8389** | **0.6292** |

**Baselines** We include 5 non-deep-learning and 6 deep-learning algorithms in our comparisons.

---

[1] http://jmcauley.ucsd.edu/data/amazon/

Table 2: Comparing SASRec, SSE-PT and SSE-PT++ on Movielens1M Dataset while varying the maximum length allowed and dimension of embeddings.

| METHODS | NDCG@10 | RECALL@10 | MAX LEN | USER DIM | ITEM DIM |
|---------|---------|-----------|---------|----------|----------|
| SASREC | 0.5769 | 0.8045 | 100 | N/A | 100 |
| SASREC | 0.5936 | 0.8233 | 200 | N/A | 50 |
| SASREC | 0.5919 | 0.8202 | 200 | N/A | 100 |
| SSE-PT | 0.6142 | 0.8212 | 100 | 50 | 100 |
| SSE-PT | 0.6191 | 0.8358 | 200 | 50 | 50 |
| SSE-PT | 0.6281 | 0.8341 | 200 | 50 | 100 |
| SSE-PT++ | 0.6186 | 0.8318 | 100 | 50 | 100 |
| SSE-PT++ | 0.6208 | 0.8358 | 200 | 50 | 50 |
| SSE-PT++ | **0.6292** | **0.8389** | 200 | 50 | 100 |

**Non-deep-learning Baselines** The simplest baseline is *PopRec*, basically ranking items according to their popularity. More advanced methods such as matrix factorization based baselines include Bayesian personalized ranking for implicit feedback (Rendle et al., 2009), namely *BPR*; Factorized Markov Chains and Personalized Factorized Markov Chains models (Rendle et al., 2010) also known as *FMC* and *PFMC*; and translation based method (He et al., 2017) called *TransRec*.

**Deep-learning Baselines** Recent years have seen many advances in deep learning for sequential recommendations. *GRU4Rec* is the first RNN-based method proposed for this problem (Hidasi et al., 2015); *GRU4Rec$^+$* (Hidasi and Karatzoglou, 2018) later is proposed to address some shortcomings of the initial version. *Caser* is the corresponding CNN-based method (Tang and Wang, 2018). *STAMP* (Liu et al., 2018) utilizes the attention mechanism without using RNN or CNN as building blocks. Very recently, *SASRec* utilizes state-of-art Transformer encoder (Vaswani et al., 2017) with self-attention mechanisms. Hierarchical gating networks, also known as *HGN* (Ma et al., 2019) are also proposed to solve this problem.

Table 3: Comparing Different Regularizations for SSE-PT on Movielen1M Dataset. NO REG stands for no regularization. PS stands for parameter sharing across all users while PS(AGE) means PS is used within each age group. SASRec is added to last row after all SSE-PT results as a baseline.

| REGULARIZATION | NDCG@5 | % GAIN | RECALL@5 | % GAIN |
|----------------|--------|--------|----------|--------|
| NO REG (BASELINE) | 0.4855 | - | 0.6500 | - |
| PS | 0.5065 | 4.3 | 0.6656 | 2.4 |
| PS (JOB) | 0.4938 | 1.7 | 0.6570 | 1.1 |
| PS (GENDER) | 0.5110 | 5.3 | 0.6672 | 2.6 |
| PS (AGE) | 0.5133 | 5.7 | 0.6743 | 3.7 |
| $l_2$ | 0.5149 | 6.0 | 0.6786 | 4.4 |
| DROPOUT | 0.5165 | 6.4 | 0.6823 | 5.0 |
| $l_2$ + DROPOUT | 0.5293 | 9.0 | 0.6921 | 6.5 |
| SSE-SE | 0.5393 | 11.1 | 0.6977 | 7.3 |
| $l_2$ + SSE-SE + DROPOUT | **0.5870** | **20.9** | **0.7442** | **14.5** |
| SASREC ($l_2$ + DROPOUT) | 0.5601 | | 0.7164 | |

**Experiment Setup** We use the same datasets as in (Kang and McAuley, 2018) and follow the same procedure in the paper: use last items for each user as test data, second-to-last as validation data and the rest as training data. We implemented our method in Tensorflow and solve it with Adam Optimizer (Kingma and Ba, 2014) with a learning rate of $0.001$, momentum exponential decay rates $\beta_1 = 0.9, \beta_2 = 0.98$ and a batch size of $128$. In Table 1, since we use the same data, the performance of previous methods except STAMP have been reported in (Kang and McAuley, 2018). We tune the dropout rate, and SSE probabilities $p_u, p_i, p_y$ for input user/item embeddings and output embeddings on validation sets and report the best NDCG and Recall for top-$K$ recommendations on test sets. For a fair comparison, we restrict all algorithms to use up to 50 hidden units for item embeddings. For the SSE-PT and SASRec models, we use the same number of transformer encoder blocks (i.e. $B = 2$) and set the maximum length $T = 200$ for Movielens 1M and 10M dataset and $T = 50$ for

other datasets. We use top-$K$ with $K = 10$ and the number of negatives $C = 100$ in the evaluation procedure. In practice, using a different $K$ and $C$ does not affect our conclusions.

**Comparisons**    One can easily see from Table 1 that our proposed SSE-PT has the best performance over all previous methods on all four datasets. On most datasets, our SSE-PT improves NDCG by more than 4% when compared with SASRec (Kang and McAuley, 2018) and more than 20% when compared to non-deep-learning methods. SSE-SE, together with dropout and weight decay, is the best choice for regularization, which is evident from Table 3. SSE-SE is a more effective way to regularize our neural networks than any existent techniques including parameter sharing, dropout, weight decay. In practice, these SSE probabilities, just like dropout rate, can be treated as tuning parameters and easily tuned. Movielens10M results are left to Table 6 in the Appendix.

| Temporal Collaborative Ranking Problem | | | |
|---|---|---|---|
| 32::Twelve Monkeys (1995)::Drama, Sci-Fi | 44::Mortal Kombat (1995)::Action, Adventure | 39::Clueless (1995)::Comedy, Romance | 11::American President, The (1995)::Comedy, Drama, Romance |
| 23::Assassins (1995)::Thriller | 42::Dead Presidents (1995)::Action, Crime, Drama | 53::Lamerica (1994)::Drama | 17::Sense and Sensibility (1995)::Drama, Romance |
| 28::Persuasion (1995)::Romance | 49::When Night Is Falling (1995)::Drama, Romance | 2::Jumanji (1995)::Adventure, Children's, Fantasy | 30::Shanghai Triad (1995)::Drama |
| 38::It Takes Two (1995)::Comedy | 19::Ace Ventura: When Nature Calls (1995)::Comedy | 14::Nixon (1995)::Drama | 34::Babe (1995)::Children's, Comedy, Drama |
| 25::Leaving Las Vegas (1995)::Drama, Romance | 12::Dracula: Dead and Loving It (1995)::Comedy, Horror | 50::Usual Suspects, The (1995)::Crime, Thriller | 41::Richard III (1995)::Drama, War |
| 37::Across the Sea of Time (1995)::Documentary | 15::Cutthroat Island (1995)::Action, Adventure, Romance | 51::Guardian Angel (1994)::Action, Drama, Thriller | 5::Father of the Bride Part II (1995)::Comedy |
| 4::Waiting to Exhale (1995)::Comedy, Drama | 43::Restoration (1995)::Drama | 16::Casino (1995)::Drama, Thriller | 31::Dangerous Minds (1995)::Drama |
| 8::Tom and Huck (1995)::Adventure, Children's | 18::Four Rooms (1995)::Thriller | 21::Get Shorty (1995)::Action, Comedy, Drama | 36::Dead Man Walking (1995)::Drama |
| 48::Pocahontas (1995)::Animation, Children's, Musical, Romance | 40::Cry, the Beloved Country (1995)::Drama | 47::Seven (Se7en) (1995)::Crime, Thriller | 33::Wings of Courage (1995)::Adventure, Romance |
| 1::Toy Story (1995)::Animation, Children's, Comedy | 46::How to Make an American Quilt (1995)::Drama, Romance | 6::Heat (1995)::Action, Crime, Thriller | 35::Carrington (1995)::Drama, Romance |
| 22::Copycat (1995)::Crime, Drama, Thriller | 27::Now and Then (1995)::Drama | 9::Sudden Death (1995)::Action | **Predict Next:** |
| 45::To Die For (1995)::Comedy, Drama | 3::Grumpier Old Men (1995)::Comedy, Romance | 13::Balto (1995)::Animation, Children's | 26::Othello (1995)::Drama |
| 10::GoldenEye (1995)::Action, Adventure, Thriller | 7::Sabrina (1995)::Comedy, Romance | 29::City of Lost Children, The (1995)::Adventure, Sci-Fi | |
| 52::Mighty Aphrodite (1995)::Comedy | 20::Money Train (1995)::Action | 24::Powder (1995)::Drama, Sci-Fi | |

| Top-5 Recommendations by SASRec | Top-5 Recommendations by Our PT |
|---|---|
| 480::Jurassic Park (1993)::Action, Adventure, Sci-Fi | 26::Othello (1995)::Drama |
| 3264::Buffy the Vampire Slayer (1992)::Comedy, Horror | 2181::Marnie (1964)::Thriller |
| 582::Metisse (1993)::Comedy | 1055::Shadow Conspiracy (1997)::Thriller |
| 3085::The Living Dead Girl (1982)::Horror | 468::The Englishman Who Went Up a Hill, But Came Down a Mountain (1995)::Comedy, Romance, Drama |
| 649::Cold Fever (1994)::Comedy, Drama | 629::Rude (1995)::Drama |

| Attention Heat Map of SASRec | Attention Heat Map of Our PT |
|---|---|
| 32 23 28 38 25 37 4 8 48 1 22 45 10 52 44 42 49 19 12 15 43 18 40 46 27 3 7 20 39 53 2 14 50 51 16 21 47 6 9 13 29 24 11 17 30 34 41 5 31 36 33 35 | 32 23 28 38 25 37 4 8 48 1 22 45 10 52 44 42 49 19 12 15 43 18 40 46 27 3 7 20 39 53 2 14 50 51 16 21 47 6 9 13 29 24 11 17 30 34 41 5 31 36 33 35 |

Figure 2: Illustration of how SASRec (Left) and SSE-PT (Right) differs on utilizing the Engagement History of A Random User in Movielens1M Dataset.

### 4.1 ATTENTION MECHANISM VISUALIZATION

Apart from evaluating our SSE-PT against SASRec using well-defined ranking metrics on real-world datasets, we also visualize the differences between both methods in terms of their attention mechanisms. In Figure 2, a random user's engagement history in Movielens1M dataset is given in temporal order (column-wise). We hide the last item whose index is 26 in test set and hope that a temporal collaborative ranking model can figure out item-26 is the one this user will watch next using only previous engagement history. One can see for a typical user; they tend to look at a different style of movies at different times. Earlier on, they watched a variety of movies, including Sci-Fi, animation, thriller, romance, horror, action, comedy and adventure. But later on, in the last two columns of Figure 2, drama and thriller are the two types they like to watch most, especially the drama type. In fact, they watched 9 drama movies out of recent 10 movies. For humans, it is natural to reason that the hidden movie should probably also be drama type. So what about the machine's reasoning?

For our SSE-PT, the hidden item indexed 26 is put in the first place among its top-5 recommendations. Intelligently, the SSE-PT recommends 3 drama movies, 2 thriller movies and mixing them up in positions. Interestingly, the top recommendation is 'Othello', which like the recently watched 'Richard III', is an adaptation of a Shakespeare play, and this dependence is reflected in the attention

weight. On the contrast, SASRec cannot provide top-5 recommendations that are personalized enough. It recommends a variety of action, Sci-Fi, comedy, horror, and drama movies but none of them match item-26. Although this user has watched all these types of movies in the past, they do not watch these anymore as one can easily tell from his recent history. Unfortunately, SASRec cannot capture this and does not provide personalized recommendations for this user by focusing more on drama and thriller movies. It is easy to see that in contrast, our SSE-PT model shares with human reasoning that more emphasis should be placed on recent movies.

## 4.2 Training Speed

In (Kang and McAuley, 2018), it has been shown that SAS-Rec is about 11 times faster than Caser and 17 times faster than GRU4Rec$^+$ and achieves much better NDCG@10 results so we did not include Caser and GRU4Rec$^+$ in our comparisons. In Figure 3, we only compare the training speeds and ranking performances among SASRec, SSE-PT and SSE-PT++ for Movielens1M dataset. Given that we added additional user embeddings into our SSE-PT model, it is expected that it will take slightly longer to train our model than un-personalized SASRec. We find empirically that training speed of the SSE-PT and SSE-PT++ model are comparable to that of SASRec, with SSE-PT++ being the fastest and the best performing model. It is clear that our SSE-PT and SSE-PT++ achieve much better ranking performances than our baseline SASRec using the same training time.

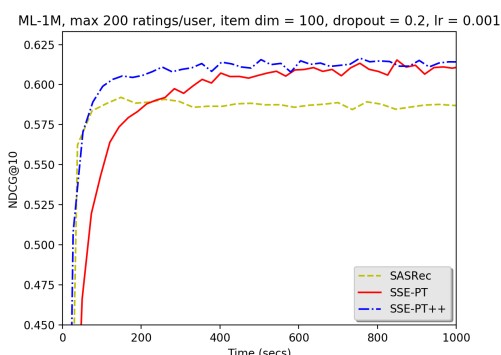

Figure 3: Illustration of the speed of SSE-PT

## 4.3 Ablation Study

**SSE probability**    Given the importance of SSE regularization for our SSE-PT model, we carefully examined the SSE probability for input user embedding in Table 7 in Appendix. We find that the appropriate hyper-parameter SSE probability is not very sensitive: anywhere between 0.4 and 1.0 gives good results, better than parameter sharing and not using SSE-SE. This is also evident based on comparison results in Table 3.

**Sampling Probability**    Recall that the sampling probability is unique to our SSE-PT++ model. We show in Table 8 in Appendix using an appropriate sampling probability like $0.2 \rightarrow 0.3$ would allow it to outperform SSE-PT when the same maximum length is used.

**Number of Attention Blocks**    We find for our SSE-PT model, a larger number of attention blocks is preferred. One can easily see in Table 9 in Appendix, the optimal ranking performances are achieved at $B = 4$ or $5$ for Movielens1M dataset and at $B = 6$ for Movielens10M dataset.

**Personalization and Number of Negatives Sampled**    Based on the results in Table 10 in Appendix, we are positive that the personalized model always outperforms the un-personalized one when we use the same regularization techniques. This holds true regardless of how many negatives sampled or what ranking metrics are used during evaluation.

## 5 Conclusion

In this paper, we propose a novel neural network architecture called Personalized Transformer for the temporal collaborative ranking problem. It enjoys the benefits of being a personalized model, therefore achieving better ranking results for individual users than the current state-of-the-art. By examining the attention mechanisms during inference, the model is also more interpretable and tends to pay more attention to recent items in long sequences than un-personalized deep learning models.

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

## 6 APPENDIX

- NDCG@$K$: defined as:

$$\text{NDCG@}K = \frac{1}{n} \sum_{i=1}^{n} \frac{\text{DCG@}K(i, \Pi_i)}{\text{DCG@}K(i, \Pi_i^*)}, \tag{6}$$

where $i$ represents $i$-th user and

$$\text{DCG@}K(i, \Pi_i) = \sum_{l=1}^{K} \frac{2^{R_{i\Pi_{il}}} - 1}{\log_2(l+1)}. \tag{7}$$

In the DCG definition, $\Pi_{il}$ represents the index of the $l$-th ranked item for user $i$ in test data based on the learned score matrix $X$. $R$ is the rating matrix and $R_{ij}$ is the rating given to item $j$ by user $i$. $\Pi_i^*$ is the ordering provided by the ground truth rating.

- Recall@$K$: defined as a fraction of positive items retrieved by the top $K$ recommendations the model makes:

$$\text{Recall@}K = \frac{\sum_{i=1}^{n} \mathbb{1}\{\exists 1 \leq l \leq K : R_{i\Pi_{il}} = 1\}}{n}, \tag{8}$$

here we already assume there is only a single positive item that user will engage next and the indicator function $\mathbb{1}\{\exists 1 \leq l \leq k : R_{i\Pi_{il}} = 1\}$ is defined to indicate whether the positive item falls into the top $K$ position in our obtained ranked list using scores predicted in (4).

**Layer Normalization**     Layer normalization (Ba et al., 2016) normalizes neurons within a layer. Previous studies (Ba et al., 2016) show it is more effective than batch normalization for training recurrent neural networks (RNNs). One alternative is the batch normalization (Ioffe and Szegedy, 2015) but we find it does not work as well as the layer normalization in practice even for a reasonable large batch size of 128. Therefore, our SSE-PT model adopts layer normalization.

**Residual Connections**     Residual connections are firstly proposed in ResNet for image classification problems (He et al., 2016). Recent research finds that residual connections can help training very deep neural networks even if they are not convolutional neural networks (Vaswani et al., 2017). Using residual connections allows us to train very deep neural networks here. For example, the best performing model for Movielens10M dataset in Table 9 is the SSE-PT with 6 attention blocks, in which $1 + 6 * 3 + 1 = 20$ layers are trained end-to-end.

**Weight Decay**     Weight decay (Krogh and Hertz, 1992), also known as $l_2$ regularization (Hoerl and Kennard, 1970), is applied to all embeddings, including both user and item embeddings.

**Dropout**     Dropout (Srivastava et al., 2014) is applied to the embedding layer $E$, self-attention layer and pointwise feed-forward layer by stochastically dropping some percentage of hidden units to prevent co-adaption of neurons. Dropout has been shown to be an effective way of regularizing deep learning models.

In summary, layer normalization and dropout are used in all layers except prediction layer. Residual connections are used in both self-attention layer and pointwise feed-forward layer. SSE-SE is used in embedding layer and prediction layer.

Table 4: Description of Datasets Used in Evaluations.

| DATASET | #USERS | #ITEMS | AVG SEQUENCE LEN | MAX SEQUENCE LEN |
|---------|--------|--------|------------------|------------------|
| BEAUTY | 52,024 | 57,289 | 7.6 | 291 |
| GAMES | 31,013 | 23,715 | 7.3 | 858 |
| STEAM | 334,730 | 13,047 | 11.0 | 1,229 |
| ML-1M | 6,040 | 3,416 | 163.5 | 2,275 |
| ML-10M | 69,878 | 65,133 | 141.1 | 7,357 |

Table 5: Comparing our SSE-PT, SSE-PT++ with SASRec on Movielen1M dataset. We use number of negatives $C = 100$, dropout probability of $0.2$ and learning rate of $1e^{-3}$ for all experiments while varying others. $p_u, p_i, p_u$ are SSE probabilities for user embedding, input item embedding and output item embedding respectively.

| | Movielens1m | | Dimensions | | Number of Blocks | Sampling Probability | SSE-SE Parameters | | |
| Model | NDCG@10 | Recall@10 | $d_u$ | $d_i$ | $b$ | $p_s$ | $p_u$ | $p_i$ | $p_y$ |
|---|---|---|---|---|---|---|---|---|---|
| SASRec | 0.5961 | 0.8195 | - | 50 | 2 | - | - | - | - |
| SASRec | 0.5941 | 0.8182 | - | 100 | 2 | - | - | - | - |
| SASRec | **0.5996** | **0.8272** | - | 100 | 6 | - | - | - | - |
| SSE-PT | 0.6101 | 0.8343 | 50 | 50 | 2 | - | 0.92 | 0.1 | 0 |
| SSE-PT | 0.6164 | 0.8336 | 50 | 50 | 2 | - | 0.92 | 0 | 0.1 |
| SSE-PT | 0.5832 | 0.8091 | 50 | 50 | 2 | - | 0 | 0.1 | 0.1 |
| SSE-PT | **0.6174** | **0.8351** | 50 | 50 | 2 | - | 0.92 | 0.1 | 0.1 |
| SSE-PT | 0.5949 | 0.8205 | 75 | 25 | 2 | - | 0.92 | 0.1 | 0.1 |
| SSE-PT | **0.6214** | **0.8359** | 25 | 75 | 2 | - | 0.92 | 0.1 | 0.1 |
| SSE-PT | 0.6281 | 0.8341 | 50 | 100 | 2 | - | 0.92 | 0.1 | 0.1 |
| SSE-PT++ | **0.6292** | **0.8389** | 50 | 100 | 2 | 0.3 | 0.92 | 0.1 | 0.1 |

Table 6: Comparing our SSE-PT with SASRec on Movielens10M dataset. Unlike Table 5, we use the number of negatives $C = 500$ instead of 100 as $C = 100$ is too easy for this dataset and it gets too difficult to tell the differences between different methods: Hit Ratio@10 approaches 1.

| | Movielens1m | | Dimensions | | Number of Blocks | SSE-SE Parameters | | |
| Model | NDCG@10 | Hit Ratio@10 | $d_u$ | $d_i$ | $b$ | $p_u$ | $p_i$ | $p_y$ |
|---|---|---|---|---|---|---|---|---|
| SASRec | 0.7268 | 0.9429 | - | 50 | 2 | - | - | - |
| SASRec | 0.7413 | 0.9474 | - | 100 | 2 | - | - | - |
| SSE-PT | 0.7199 | 0.9331 | 50 | 100 | 2 | PS | 0.01 | 0.01 |
| SSE-PT | 0.7169 | 0.9296 | 50 | 100 | 2 | 0.0 | 0.01 | 0.01 |
| SSE-PT | 0.7398 | 0.9418 | 50 | 100 | 2 | 0.2 | 0.01 | 0.01 |
| SSE-PT | 0.7500 | 0.9500 | 50 | 100 | 2 | 0.4 | 0.01 | 0.01 |
| SSE-PT | 0.7484 | 0.9480 | 50 | 100 | 2 | 0.6 | 0.01 | 0.01 |
| SSE-PT | **0.7529** | 0.9485 | 50 | 100 | 2 | 0.8 | 0.01 | 0.01 |
| SSE-PT | 0.7503 | **0.9505** | 50 | 100 | 2 | 1.0 | 0.01 | 0.01 |

- PopRec: ranking items according to their popularity.

- BPR: Bayesian personalized ranking for implicit feedback setting (Rendle et al., 2009). It is a low-rank matrix factorization model with a pairwise loss function. But it does not utilize the temporal information. Therefore, it serves as a strong baseline for non-temporal methods.

- FMC: Factorized Markov Chains: a first-order Markov Chain method, in which predictions are made only based on previously engaged item.

- PFMC: a personalized Markov chain model (Rendle et al., 2010) that combines matrix factorization and first-order Markov Chain to take advantage of both users' latent long-term preferences as well as short-term item transitions.

- TransRec: a first-order sequential recommendation method (He et al., 2017) in which items are embedded into a transition space and users are modelled as translation vectors operating on item sequences.

SQL-Rank (Wu et al., 2018) and item-based recommendations (Sarwar et al., 2001) are omitted because the former is similar to BPR (Rendle et al., 2009) except using the listwise loss function instead of the pairwise loss function and the latter has been shown inferior to TransRec (He et al., 2017).

### 6.0.1 DEEP-LEARNING BASELINES

- GRU4Rec: the first RNN-based method proposed for the session-based recommendation problem (Hidasi et al., 2015). It utilizes the GRU structures (Chung et al., 2014) initially proposed for speech modelling.

- GRU4Rec$^+$: follow-up work of GRU4Rec by the same authors: the model has a very similar architecture to GRU4Rec but has a more complicated loss function (Hidasi and Karatzoglou, 2018).
- Caser: a CNN-based method (Tang and Wang, 2018) which embeds a sequence of recent items in both time and latent spaces forming an 'image' before learning local features through horizontal and vertical convolutional filters. In (Tang and Wang, 2018), user embeddings are included in the prediction layer only. On the contrast, in our Personalized Transformer, user embeddings are also introduced in the lowest embedding layer so they can play an important role in self-attention mechanisms as well as in prediction stages.
- STAMP: a session-based recommendation algorithm (Liu et al., 2018) using attention mechanism. (Liu et al., 2018) only uses fully connected layers with one attention block that is not self-attentive.
- SASRec: a self-attentive sequential recommendation method (Kang and McAuley, 2018) motivated by Transformer in NLP (Vaswani et al., 2017). Unlike our method SSE-PT, SASRec does not incorporate user embedding and therefore is not a personalized method. SASRec paper (Kang and McAuley, 2018) also does not utilize SSE (Wu et al., 2019) for further regularization: only dropout and weight decay are used.
- HGN: hierarchical gating networks method to solve the sequential recommendation problem (Ma et al., 2019), which incorporates the user embeddings and gating networks for better personalization than the SASRec model.

Table 7: Comparing Different SSE probability for user embeddings for SSE-PT on Movielens1M Dataset. Embedding hidden units of 50 for users and 100 for items, attention blocks of 2, SSE probability of 0.01 for item embeddings, dropout probability of 0.2 and max length of 200 are used.

| USER-SIDE SSE-SE PROBABILITY | NDCG@10 | RECALL@10 |
|---|---|---|
| PARAMETER SHARING | 0.6188 | 0.8294 |
| 1.0 | 0.6258 | 0.8346 |
| 0.9 | **0.6275** | 0.8321 |
| 0.8 | 0.6244 | 0.8359 |
| 0.6 | 0.6256 | 0.8341 |
| 0.4 | 0.6237 | **0.8369** |
| 0.2 | 0.6163 | 0.8281 |
| 0.0 | 0.5908 | 0.8048 |

Table 8: Comparing Different Sampling Probability, $p_s$, of SSE-PT++ on Movielens1M Dataset. Hyper-parameters the same as Table 7, except that the max length $T$ allowed is set 100 instead of 200 to show effects of sampling sequences.

| SAMPLING PROBABILITY | NDCG@10 | RECALL@10 |
|---|---|---|
| SASREC ($T = 100$) | 0.5769 | 0.8045 |
| SSE-PT ($T = 100$) | 0.6142 | 0.8212 |
| 1.0 | 0.5697 | 0.7977 |
| 0.8 | 0.5735 | 0.7801 |
| 0.6 | 0.6062 | 0.8242 |
| 0.4 | 0.6113 | 0.8273 |
| 0.3 | 0.6186 | **0.8318** |
| 0.2 | **0.6193** | 0.8233 |
| 0.0 | 0.6142 | 0.8212 |

Table 9: Comparing Different Number of Blocks for SSE-PT while Keeping The Rest Fixed on Movielens1M and Movielens10M Datasets.

| DATASETS | # OF BLOCKS | NDCG@10 | RECALL@10 |
|---|---|---|---|
| | SASREC (6 BLOCKS) | 0.5984 | 0.8207 |
| MOVIELENS1M | 1 | 0.6162 | 0.8301 |
| | 2 | 0.6280 | 0.8365 |
| | 3 | 0.6293 | 0.8376 |
| | 4 | 0.6270 | **0.8401** |
| | 5 | **0.6308** | 0.8361 |
| | 6 | 0.6270 | 0.8397 |
| | SASREC (6 BLOCKS) | 0.7531 | 0.9490 |
| MOVIELENS10M | 1 | 0.7454 | 0.9478 |
| | 2 | 0.7512 | 0.9522 |
| | 3 | 0.7543 | 0.9491 |
| | 4 | 0.7608 | 0.9485 |
| | 5 | 0.7619 | 0.9524 |
| | 6 | **0.7683** | **0.9537** |

Table 10: Varying number of negatives $C$ in evaluation on Movielens1M dataset. Other hyper-parameters are fixed for a fair comparison.

| METRIC | NDCG@10 | RECALL@10 | $C$ |
|---|---|---|---|
| UN-PERSONALIZED | 0.3787 | 0.6119 | 500 |
| PERSONALIZED | **0.3846** | **0.6171** | 500 |
| UN-PERSONALIZED | 0.2791 | 0.4781 | 1000 |
| PERSONALIZED | **0.2860** | **0.4929** | 1000 |
| UN-PERSONALIZED | 0.1939 | 0.3515 | 2000 |
| PERSONALIZED | **0.1993** | **0.3667** | 2000 |

