# OpenReview forum: "SSE-PT: Sequential Recommendation Via Personalized Transformer"
_ICLR.cc/2020/Conference — Reject_

### Official Review · AnonReviewer2 · 2019-10-12
**Official Blind Review #2**

**Rating:** 6

**Review:**

In this paper, the authors study an important recommendation problem, i.e., sequential recommendation, and design a novel and improved model called SSE-PT (Stochastic Shared Embedding - Personalized Transformer). Specifically, the authors mainly follow the previous works of the Transformer model and the stochastic shared embedding (SSE) regularization technique. For the part of the personalized transfer (PT), the authors introduce the user embedding for each user $i$, i.e., $u_i$, shown in Eq.(2) and illustrated in Figure 1. For the part of regularization, the authors find that the SSE technique works well in terms of avoid overfiting in context of other regularization techniques.

Extensive empirical studies on five datasets show the effectiveness of the proposed approach compared with other related methods.

Overall, the paper is very well presented, in particular of the introduction and discussion about the related works, and the analysis of the experimental results.

My major concern is that the technical novelty is somehow limited in terms of the two closely related works of Transformer and stochastic shared embedding (SSE). I thus recommend weak acceptance.

Some suggestion: Some important baseline methods may be included to make the results more convincing, e.g., Fossil, MARank, and/or BERT4Rec.

Some minors:
Typo: in the paragraph below Eq.(3): user $l$ -> user $i$
Typo: FPMF, PFMC in different places


**Experience Assessment:**

I have published one or two papers in this area.

**Review Assessment: Checking Correctness Of Derivations And Theory:**

N/A

**Review Assessment: Checking Correctness Of Experiments:**

I assessed the sensibility of the experiments.

**Review Assessment: Thoroughness In Paper Reading:**

N/A

---

> ### Author Response · Authors · 2019-11-08
> **Personalization of NLP models (such as Transformer and BERT) is an important research direction**
>
> Hi Reviewer, thank you very much for your constructive and just-to-the-point feedback.
>
> While our paper does build on previous work, we think that the paper is an important contribution for 2 reasons:
> 1. In the SASRec paper, they come to the conclusion that they "empirically find that adding an explicit user embedding doesn’t improve performance (presumably because the model already considers all of the user’s actions)."  Contrary to this, we find that personalization is actually possible for transformer-based models and is proving to be very useful for recommendation systems in terms of both performance and interpretation.
>
> 2. We show that coming up with models that can incorporate long sequences should be an important research direction (our simple extension SSE-PT++ proved that).
>
> We will definitely correct the typos in our final version of the paper and will include more baselines such as Fossil, MARank, and/or BERT4Rec into our final version of the paper. We think our work is orthogonal to important works like BERT4Rec, because BERT4Rec is essentially another transformer-based approach, which may also benefit from our proposed personalization scheme. We will try to see if a similar technique to ours for SASRec also works for un-personalized models such as BERT4Rec. Even BERT4Rec authors also stated in future work section: "Another interesting direction for the future work would be introducing user component into the model for explicit user modeling when the users have multiple sessions." We think our work is first of this kind exploring this direction.

---

### Official Review · AnonReviewer3 · 2019-10-22
**Official Blind Review #3**

**Rating:** 1

**Review:**

The manuscript proposes SSE-PT, a sequential recommendation model based on transformer and stochastic shared embedding (SST). Experiments on several datasets show that SSE-PT outperforms a number of baseline methods. Some analytical results are also provided. Overall, I think this work is not suitable for ICLR due to following reasons.

The novelty of this work is limited. This work is based on SASREC [W Kang, ICDM2018] and uses transformer to encode user-item interactions in sequential manner. The difference is that this work adds user embedding in bottom layer and utilizes SSE for regularization as well as designs SSE-PT++ by sampling. To me, there is little extension or novelty.

The experiment results are not convincing. Most of results are copied from [W Kang, ICDM2018] except HGN in Table 1. Table 1 shows SASREC is much better than HGN [C Ma, KDD2019]. However, I checked the results in HGN paper and found HGN is much better than SASREC. Even though datasets are different, most of them are from Amazon data. I was not convinced by this result due to the large difference. In addition, I did not understand why the authors change evaluation metrics in Table 3, i.e., from NDCG/Recall@10 to NDCG/Recall@5. I found SSE-PT without regularization and with different regularizations are much worse than the best result, which makes me concern about the effectiveness of personalized transformer. I did not see ablation study or discussion about this.

Update: I have considered author rebuttal. I appreciate the extensive hyper-parameter sensitivity and ablation study in the paper, while these cannot be a key factor in evaluating paper as most of them can be done easily. I main concerns still lie in the novelty and experimental results. I still think this work is not suitable for ICLR and I keep my score.

**Experience Assessment:**

I have published one or two papers in this area.

**Review Assessment: Checking Correctness Of Derivations And Theory:**

N/A

**Review Assessment: Checking Correctness Of Experiments:**

I carefully checked the experiments.

**Review Assessment: Thoroughness In Paper Reading:**

I read the paper thoroughly.

---

> ### Author Response · Authors · 2019-11-08
> **Clarifying Doubts on Experimental Section**
>
> Hi Reviewer, thank you very much for raising your confusion to us on experiments. We will do a better job in clarifying on how we compared with HGN in experimental section.
>
> To explain why HGN is not doing as well as SASRec in our reported results: First, it is worth noting that the HGN paper not only used completely different datasets, but also used very distinct evaluation procedures from SASRec. Instead of predicting next engaged item, it has 10% interactions in test set, such that one prediction is correct as long as it falls into the test set, while we (and SASRec) are doing another task of predicting precisely next item, in which there is only 1 correct answer. So the task they consider is an easier task than us. Moreover, If you read the HGN paper carefully, they are mainly focused on accommodating very short sequences. In paper's experiments, they use hyper-parameter $L = 5$, where $L$ is the length of sequence used for training and inference. On contrast, our method SSE-PT and SASRec uses $L = 200$ for Movielens1m and $L=50$ for other datasets. We think that mainly accounts for the difference in original paper's reported performances and our reported performances. It is very possible that for very short sequences, HGN works quite well, better than SASRec as they have shown in their paper. We will add this delicate detail to the final version of the paper to avoid any confusions for future readers. Moreover, we modified original HGN codes to make HGN's evaluation the same as that of SASRec and open sourced at: https://github.com/SSE-PT/SSE-PT/tree/master/HGN_baseline. You have a look at our codes for both our SSE-PT and HGN baseline.
>
> As to your other comments.
> 1. Yes, it is correct that first few rows (A to F and H, I) of results in Table 1 are from SASRec paper, the reason is that we use the exactly same experimental settings on exactly same datasets. So we decided to trust the results reported in SASRec paper for older methods. We include those earlier baselines for completeness but those are not as important as SASRec because SASRec has been shown to outperform those methods. We did re-run SASRec and got slightly better results in Table 1 than the ones originally reported in SASRec paper.
>
> 2. Yes, Table 3 we use different metrics than Table 1, because we realize the NDCG@10, Recall@10 does not accurately reflect how bad over-fitting is as NDCG@5 and Recall@5. The percentages of improvement for using a well-suited regularization are much more dramatic once you switch the metrics to top 5 from top 10. This means good regularization are extremely important for top k ranking results, especially when $k$ is small. The results in Table 3 would still hold for top 10 but less dramatic for percentages of gains. On the other hand, because we want to make a fair comparison with SASRec on the same datasets, we chose to use same top-10 metrics in Table 1. This is our reasoning as to why metrics used in Table 1 and 3 are different.
>
> 3. As to the ablation study, the ablation study of personalization is done in Table 10 in Appendix and we have had a dedicated section 4.3 for different ablation studies done for each component of the model.
>
> While our paper does build on previous work, we think that the paper is an important contribution for 2 reasons:
> 1. In the SASRec paper, they come to the conclusion that they "empirically find that adding an explicit user embedding doesn’t improve performance (presumably because the model already considers all of the user’s actions)."  Contrary to this, we find that personalization is actually possible for transformer-based models and
> is proving to be very useful for recommendation systems in terms of both performance and interpretation.
>
> 2. We show that coming up with models that can incorporate long sequences should be an important research direction (our simple extension SSE-PT++ proved that).

---

### Official Review · AnonReviewer1 · 2019-10-24
**Official Blind Review #1**

**Rating:** 6

**Review:**

The paper proposes SSE-PT for sequential recommendation, which is an extension of previous work SASRec by adding user embedding with SSE  regularization [Wu et al. 2019] . They further extend SSE-PT to SSE-PT++ to handle longer sequence. Experiments on five datasets show that the SSE-PT and SSE-PT++ outperform several baseline approaches.

Detailed comments:

1)	The technical contribution seems to be scattered: user embedding is introduced, effect of different types of regularization is studied and sampling based approach is added to address long sequence. It could be better if the author could make clear what the major contribution of this paper is. Also, SSE [Wu et al. 2019] is existing technique and simply applying it to sequential recommendation is a bit incremental.

2)    In addition to SASRec, there are some other transformer based model (e.g., [1]) for sequential recommendation and the paper discuss how the proposed method differ from them.

3)	In SSE-PT++, would sampling start index v based on the recency (e.g., with exponential decay) make more sense than uniform probability?

4） Overall, experiments look comprehensive: The baseline methods include both non-deep-learning methods and recent deep learning based methods for sequential recommendation; ablation study is conducted; case study is performed on MovieLens to show how the attention weights differ from SASRec; running time is compared against baselines and sensitivity analysis on hyper-parameters are also provided.


To summarize, the paper is a bit incremental/scattered in terms of technical contribution but the execution of this paper looks solid. I would give a “weak accept” to this paper given the reasons listed above.


[1] F. Sun et. al. BERT4Rec: Sequential Recommendation with Bidirectional Encoder Representations from Transformer


**Experience Assessment:**

I have published one or two papers in this area.

**Review Assessment: Checking Correctness Of Derivations And Theory:**

I assessed the sensibility of the derivations and theory.

**Review Assessment: Checking Correctness Of Experiments:**

I assessed the sensibility of the experiments.

**Review Assessment: Thoroughness In Paper Reading:**

I read the paper at least twice and used my best judgement in assessing the paper.

---

> ### Author Response · Authors · 2019-11-08
> **Exponential decay idea does work empirically better than Uniform one**
>
> Hi Reviewer, Thank you very much for your insightful feedback and suggestions.
>
> While our paper does build on previous work, we think that the paper is an important contribution for 2 reasons:
> 1. In the SASRec paper, they come to the conclusion that they "empirically find that adding an explicit user embedding doesn’t improve performance (presumably because the model already considers all of the user’s actions)."  Contrary to this, we find that personalization is actually possible for transformer-based models and is proving to be very useful for recommendation systems in terms of both performance and interpretation.
>
> 2. We show that coming up with models that can incorporate long sequences should be an important research direction (our simple extension SSE-PT++ proved that).
>
> Yes, we should definitely include CIKM'19 BERT4Rec in the final version of the paper, which we were not aware of. We think our work is orthogonal to important works like BERT4Rec, because BERT4Rec is essentially another transformer-based approach, which may also benefit from our proposed personalization scheme. We will try to see if a similar technique to ours for SASRec also works for un-personalized models such as BERT4Rec. Even BERT4Rec authors also stated in future work section: "Another interesting direction for the future work would be introducing user component into the model for explicit user modeling when the users have multiple sessions." We think our work is first of this kind exploring this direction.
>
> Also, your idea of sampling start index $v$ based on the recency (e.g., with exponential decay) sounds very intuitive and could be very promising. We did a quick experiment, we find using exponential decay gives slightly better results on movielen1m data when we use max length of 50. We find using combination of our idea and your idea (most of weight on the last start index but rest of times we sample start index based on recency with exponential decay) empirically performs better, giving NDCG@10 of 0.59945 versus 0.59509 and Recall@10 of 0.81109 versus 0.80414. I think our works point to a future direction that worth more explorations, both empirically and theoretically.

---

### Decision · Program_Chairs · 2019-12-19

**Decision:**

Reject

**Comment:**

The paper proposes to improve sequential recommendation by extending SASRec (from prior work) by adding user embedding with SSE regularization.  The authors show that the proposed method outperforms several baselines on five datasets.

The paper received two weak accepts and one reject.  Reviewers expressed concerns about the limited/scattered technical contribution.  Reviewers were also concerned about the quality of the experiment results and need to compare against more baselines.  After examining some related work, the AC agrees with the reviewers that there is also many recent relevant work such as BERT4Rec that should be cited and discussed.  It would make the paper stronger if the authors can demonstrate that adding the user embedding to another method such as BERT4Rec can improve the performance of that model.  Regarding R3's concerns about the comparison against HGN, the authors indicates there are differences in the length of sequences considered and that some method may work better for shorter sequences while their method works better for longer sequences.  These details seems important to include in the paper.

In the AC's opinion, the paper quality is borderline and the work is of limited interest to the ICLR community.  Such would would be more appreciated in the recommender systems community.  The authors are encouraged to improve the paper with improved discussion of more recent work such as BERT4Rec, add comparisons against these more recent work, incorporate various suggestions from the reviewers, and resubmit to an appropriate venue.